# ERK8 is a negative regulator of O-GalNAc glycosylation and cell migration

Joanne Chia[1], Keit Min Tham[1], David James Gill[1], Emilie Anne Bard-Chapeau[1†], Frederic A Bard[1,2]*

[1]Institute of Molecular and Cell Biology, Singapore, Singapore; [2]Department of Biochemistry, National University of Singapore, Singapore, Singapore

**Abstract** ER O-glycosylation can be induced through relocalisation GalNAc-Transferases from the Golgi. This process markedly stimulates cell migration and is constitutively activated in more than 60% of breast carcinomas. How this activation is achieved remains unclear. Here, we screened 948 signalling genes using RNAi and imaging. We identified 12 negative regulators of O-glycosylation that all control GalNAc-T sub-cellular localisation. ERK8, an atypical MAPK with high basal kinase activity, is a strong hit and is partially localised at the Golgi. Its inhibition induces the relocation of GalNAc-Ts, but not of KDEL receptors, revealing the existence of two separate COPI-dependent pathways. ERK8 down-regulation, in turn, activates cell motility. In human breast and lung carcinomas, ERK8 expression is reduced while ER O-glycosylation initiation is hyperactivated. In sum, ERK8 appears as a constitutive brake on GalNAc-T relocalisation, and the loss of its expression could drive cancer aggressivity through increased cell motility.

*For correspondence:
fbard@imcb.a-star.edu.sg

†Present address: Novartis Institutes for BioMedical Research, Basel, Switzerland

Competing interests: The authors declare that no competing interests exist.

## Introduction

GalNAc-type O-linked glycans are polysaccharides present on secreted and membrane-inserted proteins (*Tran and Ten Hagen, 2013*). Traditionally associated with mucin-like proteins, recent advances in mass spectrometric analysis have revealed O-glycosylation on hundreds of different proteins (*Steentoft et al., 2013*). Recent results have also highlighted specific functional roles for O-glycans; for instance, in regulating the secretion of the phosphatemia regulator FGF23 (*Kato et al., 2006*) and in processing of the angiopoietin-like factor ANGPTL2 (*Schjoldager et al., 2010*).

O-glycans are synthesised through the step-wise action of various glycosylation enzymes, starting with the UDP-N-Acetyl-Alpha-D-Galactosamine:Polypeptide N-Acetyl-galactosaminyltransferases (GalNAc-Ts), a large family of 20 different isoforms that catalyses the addition of N-Acetylgalactosamine (GalNAc) onto serine or threonine residues (*Bennett et al., 2012*). The addition of GalNAc on proteins generates the Tn antigen, with antigenicity being lost upon the addition of other sugar residues. Earlier work demonstrated that carcinomas stain prominently with antibodies and lectins such as the *Helix Pomatia* Lectin (HPL), which binds to Tn antigens (*Springer, 1983*). The high prevalence and specificity of this cancer glycophenotype is remarkable, with matching normal tissues and benign tumours expressing much lower levels.

This increase in Tn levels is proposed to stem from a block or reduction in the activity of the main O-GalNAc-modifying enzyme, the Core 1 Galactosyl-Transferase (C1GALT) (*Ju et al., 2002a, 2008b; Stanley, 2011*); indeed, the loss of C1GALT in the high Tn-expressing T cell leukaemia Jurkat cell line has been reported (*Ju et al., 2008a*). In breast carcinoma, however, high Tn levels seem to be caused by a different mechanism: GalNAc-Ts are massively relocated from the Golgi apparatus to the endoplasmic reticulum (ER) with Tn staining largely located in the ER (*Gill et al., 2013*). Further, in some cancer cells, O-glycosylation initiation in the ER has also been reported (*Egea et al., 1993*).

**eLife digest** The likelihood of an individual being able to recover from cancer depends on: where the cancer is within the body, how quickly the disease is detected and how quickly treatment is started. Cancers that have spread from their original location to another part of the body are particular challenging to treat, and cause the vast majority of cancer deaths every year.

Treatments that can recognize and eradicate cancer cells, while leaving nearby healthy cells untouched, are still needed—and so there has been a lot of research into identifying the key differences between healthy cells and cancer cells. For several decades, researchers have been aware that cancer cells have more proteins coated with modified sugars on their cell surfaces than healthy cells. This is caused by the enzymes that add these sugars to the proteins relocating from one location within the cell, the Golgi apparatus, to another, called the endoplasmic reticulum. These specific 'sugar-coated' proteins are known to encourage cancer cells to migrate and invade new tissues, but the mechanisms that regulate the addition of these sugar molecules to proteins remains poorly understood.

Now Chia et al. have discovered 12 molecules that regulate this process, including an enzyme called ERK8 that is found at the Golgi apparatus. ERK8 is shown to prevent the relocation of the sugar-adding enzymes from the Golgi to the endoplasmic reticulum, thereby restricting the production of sugar-coated proteins that help the cancer cells to spread within the body. By identifying 12 potential targets for new therapeutics aimed at preventing the spread of cancer, the work of Chia et al. could ultimately help to improve the chances of patients recovering from certain cancers.

Trafficking of GalNAc-Ts to the ER can be stimulated by growth factors such as epidermal growth factor (EGF) and platelet-derived growth factor (PDGF), with GalNAc-Ts active in the ER and GalNAc incorporation in proteins increasing after relocation (*Gill et al., 2010*). It is surmised that glycosylation of ER-resident proteins likely explains this observed increase in Tn staining, as several of these proteins bear O-GalNAc in mass spectrometric analyses (*Steentoft et al., 2013*).

Although it is unclear which specific proteins are modified, O-glycosylation in the ER results in a marked stimulatory effect on cell adhesion and cell migration (*Gill et al., 2013*). This suggests that ER O-glycosylation promotes the invasive and metastatic potential of malignant tumour cells. Tn levels are consistently higher in higher grade, more aggressive breast tumours. Conversely, ER-specific inhibition of O-glycosylation reduced drastically lung metastasis in a mice model (*Gill et al., 2013*).

GalNAc-Ts transport is stimulated by activated SRC tyrosine kinases and requires the COPI coat (*Gill et al., 2010*). COPI is a multimeric protein complex required for the formation of transport carriers and functions in the retrograde traffic between the Golgi and the ER (*Beck et al., 2009*; *Szul and Sztul, 2011*). COPI coat assembly is regulated by small GTPases of the Arf family and their regulator, the GTP exchange factor, GBF1; however, the regulation of COPI-coated carrier formation in response to extracellular signals is poorly understood.

To better understand the mechanisms regulating Tn expression in cancer, we performed an RNAi screen targeting 948 genes presumed to be involved in signal transduction. We identified and validated 12 regulators, with a particular focus on the MAP kinase ERK8 (alias *MAPK15*), the most recently identified member of the MAPK family (*Abe et al., 2002*). Unlike classical MAP kinases, ERK8 possesses an atypically long C-terminal domain and appears to constitutively auto-phosphorylate its Thr-X-Tyr motif (*Klevernic et al., 2006*). Here, we find that a fraction of the ERK8 protein is localised at the Golgi where it specifically inhibits COPI vesicle formation and the export of GalNAc-Ts. The loss of ERK8 activity results in increased O-glycosylation and increased cell motility. We find that ERK8 expression is also frequently downregulated in lung carcinomas, which may partly explain the high Tn phenotype and invasiveness of these tumours.

## Results

### RNAi screening identifies 12 signalling genes that negatively regulate Tn levels

We recently reported the results from a screen for regulators of Golgi morphology and organisation using various markers including fluorescently labelled HPL (*Chia et al., 2012*). In the analysis presented

in this study, we quantified Tn levels using HPL fluorescence intensity per cell (*Figure 1A*). In a pilot screen using HeLa cells, which targeted 63 known players of membrane traffic (*Chia et al., 2012*), we identified that knockdown of the SNARE gene Syntaxin 5 (*STX5*) reproducibly induced a 6–7-fold increase in Tn levels relative to a non-targeting (NT) control (GFP siRNA) (*Figure 1A*). This effect was presumably due to a defect in the balance in anterograde vs retrograde ER-to-Golgi traffic of GalNAc-Ts. Using *STX5* and GFP siRNA as positive and negative controls, we then screened 948 signalling genes in search for regulators of O-glycosylation. We discarded the results for 134 siRNA pools that reduced the cell number to less than 20% of the control (*Figure 1B*). Of the remaining siRNA pools, we identified numerous gene knockdowns that increased HPL levels significantly more than *STX5* depletion. None of the gene depletions seemed to significantly reduce the basal levels of Tn in HeLa cells (*Figure 1C*). The knockdown effects were reproduced in independent replicates (*Figure 1–figure supplement 1A*), and the trend was mostly independent of the analysis algorithm used, although the fold increase was higher with one method than with the other (*Figure 1—figure supplement 1B*).

To focus our analysis, we used a stringent cut-off of a ninefold increase in HPL staining intensity, which resulted in 19 genes (*Figure 1B,C*). Depletion of one of these negative regulators—the Extracellular Signal Regulated Kinase 8 (ERK8)—induced a particularly marked increase in Tn, ranging from 4–25-fold depending on the experimental design and RNAi reagent (*Figure 1D*, *Figure 1—figure supplement 1C*, *Figure 3—figure supplement 1A*). It should be noted that knock-down was not optimised for the other 18 genes and thus Tn levels may reflect partly the extent of target depletion.

To exclude the possibility of off-target effects in the 19 hits, we repeated the screen using the individual siRNAs that were used in each siRNA pool in the primary screen. For ERK8, we observed that three out of the four single siRNAs significantly increased Tn levels above the NT control (*Figure 1—figure supplement 1C,D*). Using a threshold of 4.5-fold increase for at least two independent siRNAs, 12 genes were considered validated (*Figure 1—figure supplement 1D*). To verify that the effects observed were not specific to the detection method used, knockdown cells were also stained with a different lectin, *Vicia Villosa* Lectin (VVL), which revealed a highly consistent pattern (*Figure 1—figure supplement 1E*).

To validate that these genes indeed led to the up-regulation in GalNAc protein O-glycosylation, we sought to downregulate the responsible enzymes. Although GalNAc-Ts represent a large family, the T1 and T2 isoforms are by far the most prevalent and represent most of GalNAc-T activity in HeLa cells (*Bennett et al., 2012*). This is apparent in the almost complete loss of Tn levels when GalNAc-T1 and -T2 were depleted (*Figure 1E*). Co-depletion of the two enzymes also reduced significantly the Tn increase from the knockdown of ERK8 as well as that from other Tn regulators (*Figure 1E*, *Figure 1—figure supplement 1F*). This effect was not caused by inefficient ERK8 knockdown, as other co-knockdown experiments did not have such an effect. In addition, ERK8 levels were still significantly reduced in the triple knockdown configuration (*Figure 1—figure supplement 1G*).

Overall, our screen revealed several negative regulators of the O-GalNAc glycosylation process, which thus appears to be tightly controlled by signalling mechanisms.

## Tn negative regulators are not required for O-glycan extension

Two mechanisms are known to increase Tn levels: inhibition of O-GalNAc extension (*Ju et al., 2008a*, *2008b*) or relocation of GalNAc-Ts from the Golgi apparatus to the ER. The loss of expression of C1GALT or its molecular chaperone, COSMC, results in a failure to generate the subsequent Core 1 glycan structure (the TF antigen) and thus inhibits O-GalNAc extension, which is detectable as a loss in Peanut Agglutinin (PNA) lectin positive staining (*Swamy et al., 1991*). Relocation of GalNAc-Ts from the Golgi apparatus to the ER, on the other hand, induces a modest but measurable increase in PNA staining (*Gill et al., 2010*).

To distinguish between the two possibilities, PNA staining was quantified upon depletion of each of the 12 Tn regulators. HeLa cells with a stable COSMC knockout which prohibits C1GALT activity, was used as a positive control and, as expected, completely abolished PNA staining. Comparatively, there was either no significant decrease or some increase in PNA staining following depletion of each of the Tn regulators (*Figure 2B*, *Figure 2—figure supplement 1A*), with the most significant increase in PNA staining observed after ERK8 knockdown (*Figure 2A*). Therefore, none of the Tn-regulating genes we identified appears to be required for core 1-forming activity and thus do not regulate Tn levels by inhibiting O-GalNAc extension.

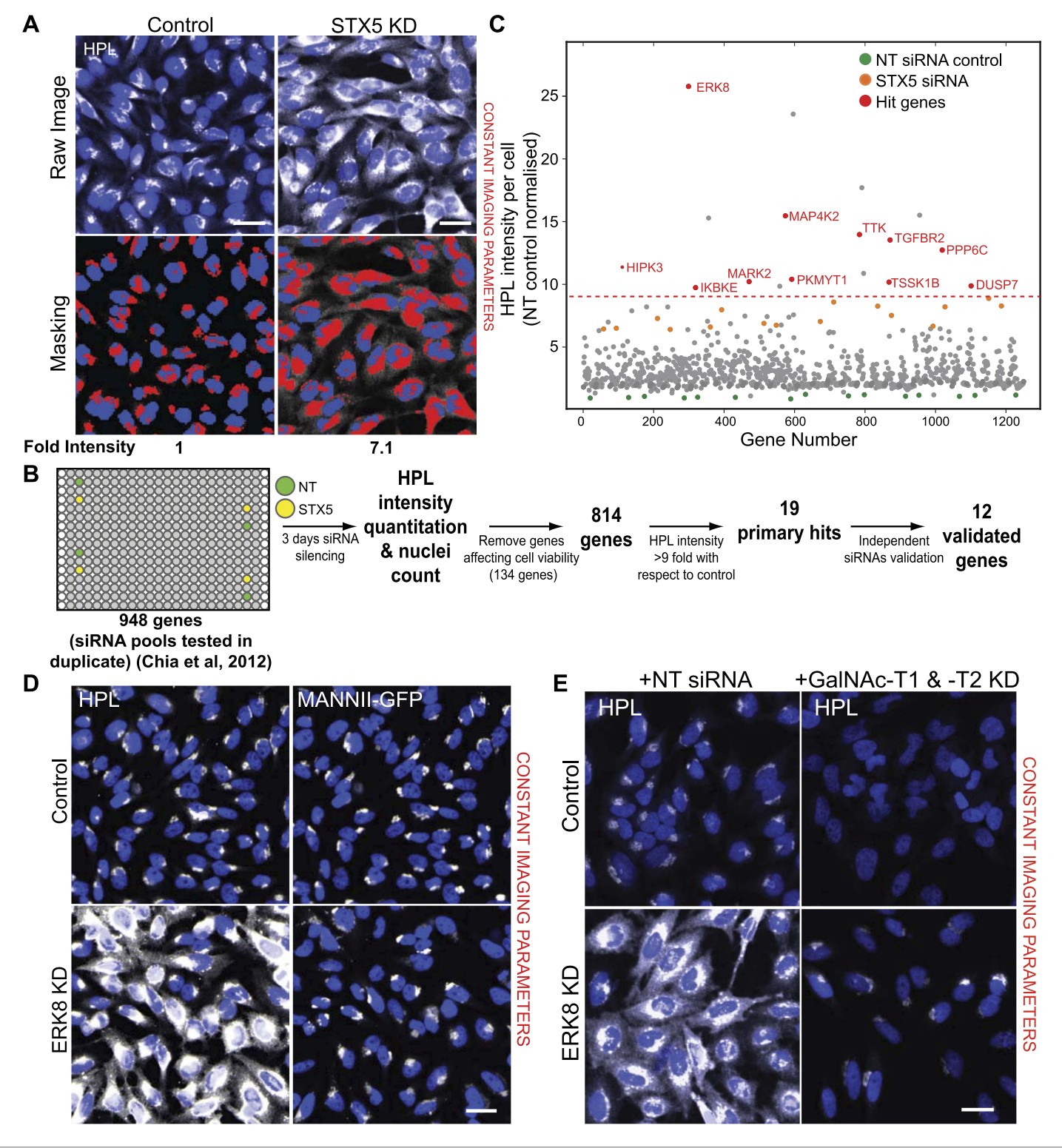

**Figure 1**. RNAi screening reveals 12 negative regulators of Tn expression. (**A**) *Helix pomatia* lectin (HPL) staining was analysed using the 'Transfluor HT' module of MetaXpress software (Molecular Devices). A mask was generated for both HPL and nuclei (Hoechst) staining to classify the region of measurement (lower panels). Scale bar: 30 µm. (**B**) Schematic overview of the screening process. Images from the RNAi screen in ***Chia et al. (2012)*** were quantified for HPL intensities. Non-targeting (NT) siRNA and Syntaxin 5 (*STX5*) siRNA were used as negative and positive controls, resepectively. (**C**) Fold-change of HPL intensities normalised to NT siRNA treatment (green dots) and *STX5* (orange dots). Primary hits were selected based on a threshold of a

*Figure 1. Continued on next page*

*Figure 1. Continued*

nine-fold increase (red dashed line) and the final validated genes are labelled in red (Hit genes). (**D**) Images from the screen of HPL staining in HeLa cells depleted of ERK8. MannII-GFP labels the Golgi apparatus. Scale bar: 30 µm. (**E**) HPL staining in cells knockdown of ERK8 with a control siRNA or GalNAc-T1 and -T2 siRNA. Scale bar: 30 µm.

The following figure supplements are available for figure 1:

**Figure supplement 1**. *Helix Pomatia* Lectin (HPL) stains reliably and specifically for Tn antigen.

## Tn levels depend on GalNAc-Ts subcellular localisation

To evaluate whether the alternative mechanism of GalNAc-Ts relocation to the ER was implicated, the subcellular distribution of GalNAc-Ts was evaluated by immunofluorescence. In control cells, GalNAc-T1 localised exclusively in the perinuclear region, co-localising with the Golgi marker MannII (*Figure 2C*). Upon ERK8 depletion, GalNAc-T1 distribution appeared more diffuse, co-localising with the ER marker Calreticulin (*Figure 2C*), with MannII-GFP staining remaining mostly perinuclear (*Figure 2C*). The MannII-GFP-positive structures were more fragmented in the knockdown cells than in the untreated cells; this finding is reminiscent of the effects of SRC activation (*Gill et al., 2010*).

Depletion of phosphatidylinositol (PI) 4-kinase (PI4KA), another major regulator of Tn, also resulted in GalNAc-T1 redistribution (*Figure 2C*). However, unlike ERK8 depletion, MannII-GFP appeared to also redistribute to a cytoplasmic pattern upon PI4KA depletion (*Figure 2C*). This suggests that the entire content of the Golgi apparatus becomes redistributed to the ER, and is consistent with our previous findings, where PI4KA depletion induced a redistribution of TGN46, a *trans* golgi marker (*Chia et al., 2012*) (*Figure 2—figure supplement 1D*).

Next, we used a quantitative approach to determine the extent of ER relocalisation that occurs in response to depletion of the other Tn-regulating genes, measuring the degree of co-localisation between GalNAc-T1 and Calreticulin staining. The Pearson's correlation coefficient between these two markers was significantly increased following the knockdown of all 12 genes (*Figure 2D*, *Figure 2—figure supplement 1B*). Most exhibited levels similar to that induced by Brefeldin A (BFA) treatment, which redistributes Golgi proteins to the ER (*Fujiwara et al., 1988*). Comparatively, there was only slight perturbation of GalNAc-T1 localisation in the COSMC knockout cells. Similar trends were also observed for GalNAc-T2 staining (*Figure 2—figure supplement 1C*). In contrast to the GalNAc-T staining, none of the signalling genes significantly affected the MannII-GFP distribution, apart from PI4KA depletion (*Figure 2—figure supplement 1E*).

Collectively, our results suggest that signalling genes influence Tn levels through the subcellular distribution of GalNAc-Ts and that, with the exception of PI4KA, they affect the trafficking of these enzymes specifically.

## Tn-regulating proteins likely act at the level of the Golgi apparatus

To explore how the products of these genes might be functioning, we retrieved data pertaining to their subcellular localisation and protein–protein interactions from Protein Atlas, GeneCards and STRING (*Jensen et al., 2009*; *Safran et al., 2010*; *Uhlen et al., 2010*). Three proteins—PI4KA, PKMYT1 and MAP4K2—have previously been reported to be localised at least partially at the Golgi apparatus (*Nakagawa, 1996*; *Ren et al., 1996*; *Liu et al., 1997*). PI4KA is proposed to generate phosphoinositol-4-phosphate (PI4P), which is essential for recruiting membrane trafficking effectors to the Golgi (*De Matteis et al., 2005*), including Vps74/GOLPH3, which retains various glycosyltransferases at the Golgi through retrograde trafficking (*Wood et al., 2009*). Although GalNAc-Ts have not been known to be regulated by Vps74/GOLPH3, it represents a potential mechanism for their retention at the Golgi. PKMYT1 is required for the reassembly of the Golgi during telophase (*Nakajima et al., 2008*). In addition, ERK8 has been reported to localise perinuclearly in A431 cells (*Uhlen et al., 2010*), suggesting a potential Golgi localisation.

Four other kinases—HIPK3, TTK, MARK2 and DUSP7— interact with Golgi-associated proteins (*Dowd et al., 1998*; *Colland et al., 2004*; *Dou et al., 2004*; *Sowa et al., 2009*; *Cui et al., 2010*). HIPK3 was also found to interact with Golgi structural protein GRASP65, the Golgi-localised LIM kinase, and the ERK8 interactor, HIC-5 (*Colland et al., 2004*). HIPK3 also interacts with PKMYT1 (*Wells et al., 1999*). The MARK2 protein controls microtubule stability through phosphorylation of

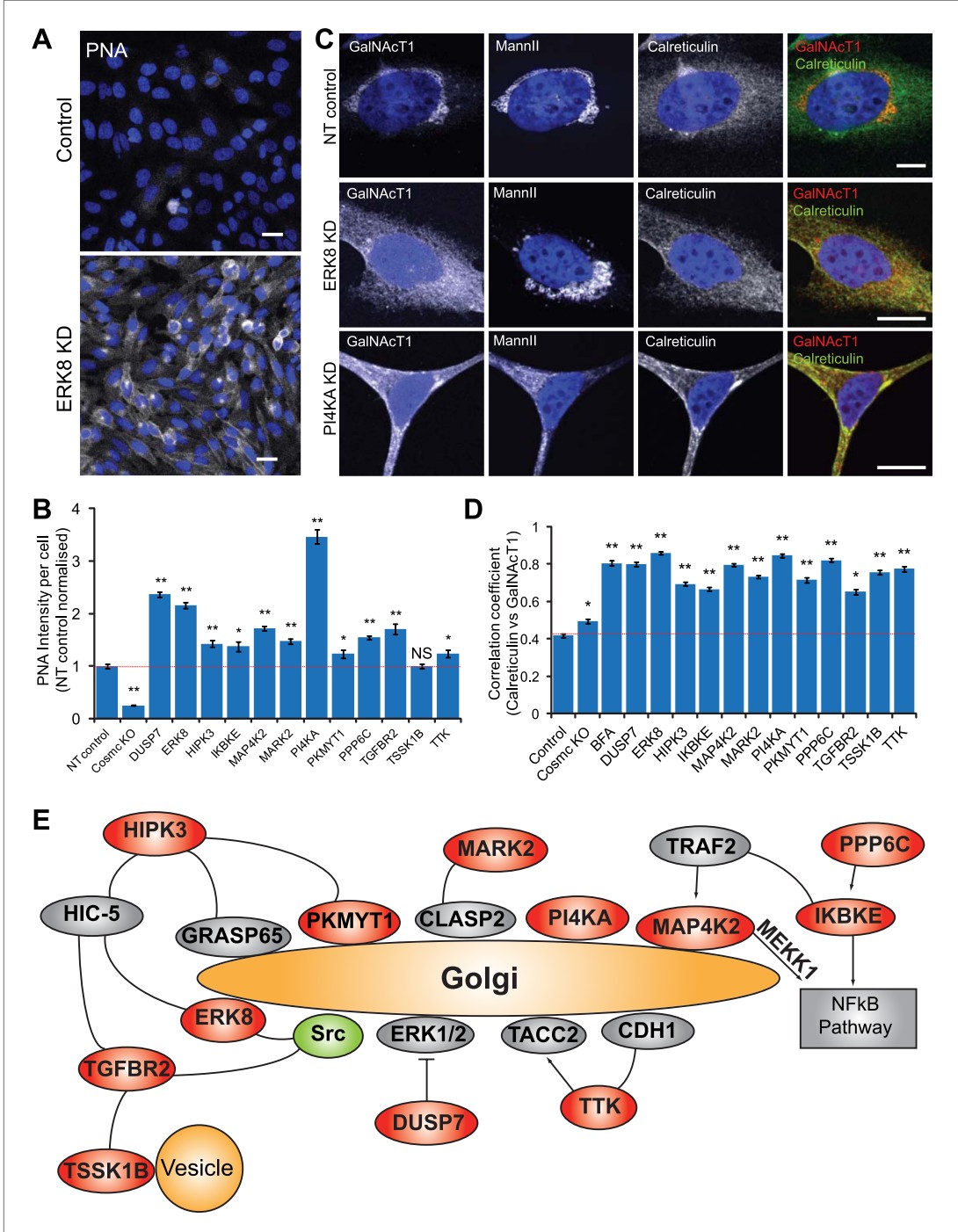

**Figure 2**. Tn regulators control Tn expression through GalNAc-T subcellular localisation. (**A**) Peanut Agglutinin (PNA) lectin staining in ERK8-depleted HeLa cells. Scale bar: 30 µm. (**B**) PNA lectin staining quantification after depletion of the 12 Tn regulators, using COSMC knockout HeLa cells as a positive control. (**C**) Co-staining for endogenous GalNAc-T1 and Golgi (MannII-GFP) and ER (Calreticulin) markers. Scale bar: 10 µm. (**D**) Co-localisation of the GalNAc-T1 and Calreticulin measured using Pearson's correlation coefficient of the staining intensities of the two markers. Cells were analysed using MetaXpress Translocation-Enhanced analysis module. Values on graphs indicate the mean ± SEM. **$p<0.0001$, *$p<0.05$ by two-tailed unpaired $t$ test, relative to NT siRNA-treated cells. (**E**) A potential regulatory network of signalling proteins regulating GalNAc-T localisation.

The following figure supplements are available for figure 2:

**Figure supplement 1**. Tn regulators control GalNAc-T1 and -T2 localisation.

microtubule-associated proteins (*Yoshimura and Miki, 2011*) and its interaction with the microtubule tracking protein, CLASP2 (*Sowa et al., 2009*), suggests CLASP2 as a potential substrate. CLASP2 is involved in microtubule nucleation at the Golgi (*Miller et al., 2009*) and microtubules could be nucleated at the *cis* Golgi (*Rivero et al., 2009*). Since Golgi-to-ER retrograde traffic depends on microtubule tracks (*Palmer et al., 2005*; *Spang, 2013*), GalNAc-T relocation could depend on CLASP2-associated microtubules regulated by MARK2.

Two proteins—ERK8 and TGFBR2—interact with SRC (*Abe et al., 2002*; *Galliher and Schiemann, 2007*). ERK8 activity was reported to increase in the presence of active SRC (*Abe et al., 2002*) and TGFBR2 is phosphorylated by SRC (*Galliher and Schiemann, 2007*). In addition, MAP4K2, IKBKE and PPP6C are linked to the canonical NFkB pathway, suggesting that this pathway might control GalNAc-T localisation (*Shimada et al., 1999*; *Chadee et al., 2002*; *Eddy et al., 2005*; *Stefansson and Brautigan, 2006*). Finally, several other interactions, either direct or with one intermediate, were found between the Tn-regulating genes.

Altogether, this analysis suggests that the Tn-regulating genes are acting at the Golgi level, perhaps part of a regulatory network controlling the subcellular localisation of GalNAc-Ts (*Figure 2E*). Further experiments are required to confirm the reality of this network and its precise connectivity.

## ERK8 kinase activity is required for O-glycosylation regulation

We next sought to understand the mechanistic basis of Tn level regulation by ERK8. ERK8 displays high basal activity in resting cells and is not stimulated by growth factor activity (*Abe et al., 2002*; *Klevernic et al., 2006*) but through auto-phosphorylation on residues Thr 175 and Tyr 177. To test if its kinase activity is important, we selected the siRNA dERK8-4 for its potency (*Figure 3—figure supplement 1B*) and designed an siRNA-resistant ERK8 construct tagged with GFP (GFP-ERK8-siR), as well as a kinase-inactive mutant counterpart (GFP-ERK8-siR-T175A-Y177F).

ERK8-depleted cells were then transfected with either wild-type or kinase-mutant ERK8 constructs 48 hr after siRNA treatment. HPL intensities were quantified for transfected (GFP-expressing) and non-transfected (non-GFP-expressing) cell populations, each comprising hundreds of cells. Non-transfected ERK8-depleted cells displayed a marked increase in HPL staining, whereas significantly lower HPL staining was observed in cells transfected with the wild-type GFP-ERK8-siR construct (*Figure 3A,B*). Importantly, HPL levels remained almost similar to non-GFP-expressing ERK8-depleted cells when the cells were transfected with the kinase-inactive mutant (GFP-ERK8-siR-T175A-Y177F) indicating that kinase activity is important for the negative regulation of O-glycosylation (*Figure 3B*).

## ERK8 inhibitor induces a rapid increase in Tn levels

Ro-31-8220 inhibits the kinase activity of ERK8 (*Klevernic et al., 2006*). To see if this compound could recapitulate the effects of ERK8 depletion, cells were treated with 5 µM Ro-31-8220 for various durations. A significant increase in Tn was observed as early as 1 hr after treatment and peaked at a nearly eightfold increase after 3.5 hr as compared with basal levels (*Figure 3C,D*). After that, we observed some cell death, which possibly explains the accompanying decrease in Tn levels.

This result indicated that the increase in Tn staining is a relatively rapid phenomenon and that O-GalNAc glycosylated protein accumulation can be achieved in a few hours. Our findings also suggest that these changes were not caused by expression changes of the O-glycoproteins or the O-glycosylation machinery. Indeed, co-treatment of cells with Ro-31-8220 and a transcriptional inhibitor α-amanitin (*Chafin et al., 1995*) did not reduce Tn levels (*Figure 3—figure supplement 1D*) and protein levels of the enzymes and chaperones involved in the early O-glycosylation stages were unaffected in ERK8-depleted cells (*Figure 3—figure supplement 1E*). The rapid effect of Ro-31-8220 suggests that ERK8 acts relatively directly on GalNAc-T traffic.

## O-glycosylation is initiated in the ER and several proteins are hyperglycosylated following ERK8 depletion

GalNAc-T staining upon ERK8 depletion strongly suggests the relocalisation of these enzymes to the ER. To further confirm this, we used an ER-specific glycosylation reporter (GFP-Muc-PTS), which contains a Pro-Thr-Ser (PTS)-rich sequence with up to 15 sites for GalNAc addition (*Gill et al., 2010*). After pulldown with HPL-conjugated beads, we found a significant increase in the glycosylation of this reporter upon treatment with Ro-31-8220 for 3.5 hr (*Figure 3E*). We also verified ER localisation using a stably expressed ER-localised GalNAc-T inhibitor described previously (*Gill et al., 2013*).

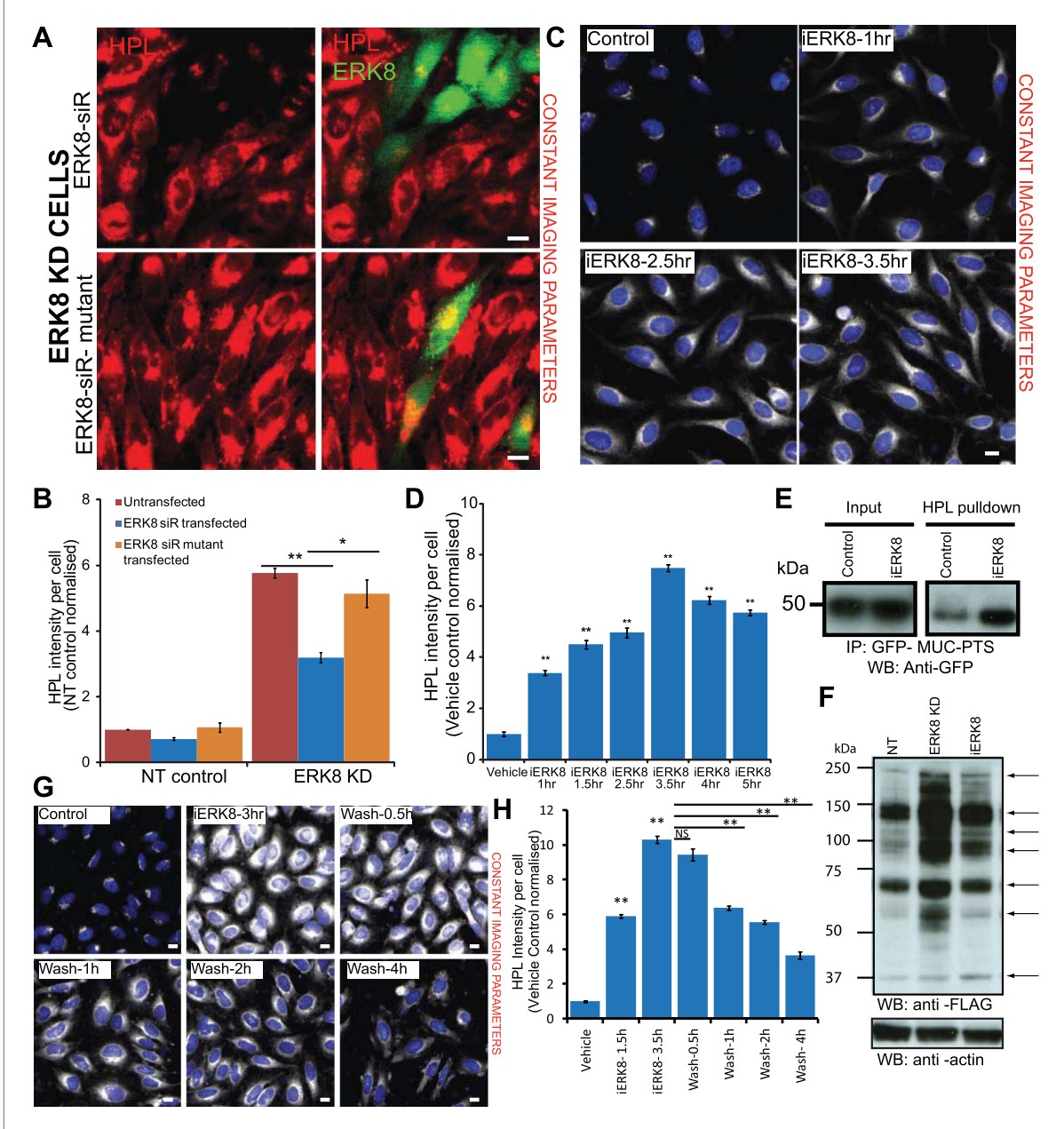

**Figure 3**. ERK8 regulates ER-localised O-glycosylation initiation. (**A**) Protein replacement by expression of siRNA-resistant wild-type ERK8 or the kinase inactive mutant in ERK8-depleted HeLa cells. Cells were stained for *Helix pomatia* lectin (HPL) and ERK8. Scale bar: 30 μm. (**B**) Tn levels of non-targeting (NT) siRNA-treated and ERK8-depleted cells that were untransfected (red bars) or transfected with wild-type ERK8 (blue bars) or kinase-inactive mutant ERK8 (orange bars). (**C**) Treatment with 5 μM ERK8 inhibitor Ro-31-8220 (iERK8) over time and staining for Tn expression with HPL in HeLa cells. (**D**) Quantification of Tn expression after 5 μM iERK8 treatment. (**E**) SDS-PAGE analysis of ER-specific glycosylation reporter (Muc-PTS) expressed in HEK293T cells treated with vehicle or with 5 μM iERK8 for 3.5 hr. Muc-PTS was immunoprecipitated using HPL-conjugated agarose. (**F**) SDS-PAGE analysis of untreated, ERK8-depleted and inhibitor-treated cell lysates metabolically labelled using GalNAz-FLAG. Arrows point to bands with changed intensities (**G**) Tn staining after 5 μM iERK8 treatment for 3 hr followed by chase over time. Scale bar: 30 μm. (**H**) Quantification of Tn expression levels upon iERK8 treatment and washout. Values on graphs indicate the mean ± SEM. \*\*p<0.0001, \*p<0.05 by two-tailed unpaired *t* test, relative to untransfected or ERK8 wild-type transfected cells in (**B**) and vehicle treated cells in (**D**) and (**H**).

The following figure supplements are available for figure 3:

**Figure supplement 1**. ERK8 regulates Tn expression through GalNAc-T relocation and not related to transcriptional or translational events.

This inhibitor counteracted the increase in Tn levels observed upon ERK8 knockdown (*Figure 3—figure supplement 1C*).

Recently, several ER-resident proteins were shown to be O-glycosylated (*Steentoft et al., 2013*). To determine the extent to which the proteome is modified upon ERK8 depletion, we metabolically labelled cells with a GalNAc sugar analogue, FLAG-GalNAz (*Laughlin and Bertozzi, 2007*). After 24 hr, cells depleted of ERK8 (by siRNA or Ro-31-8220) exhibited substantial increases in GalNAz incorporation, as revealed by the presence of several bands on SDS-PAGE gels (*Figure 3F*). This suggests that ERK8 controls the O-glycosylation status of several proteins, which probably includes ER residents.

## GalNAc-Ts relocation is rapidly reversible

The effects of Ro-31-8220 offered the possibility to test how quickly Tn levels can return to baseline levels after drug washout. High Tn levels obtained after 3.5 hr of treatment decreased progressively with an approximately 2.5-hr half-life (*Figure 3G,H*). Consistently, staining for GalNAc-Ts showed a similar trend, with a significant increase in enzyme localisation at the Golgi within 2 hr of washout (*Figure 3—figure supplement 1F,G*). This suggests that reactivation of ERK8 slows the continuous retrograde flow of GalNAc-Ts and that anterograde traffic shifts their distribution back to the Golgi. These results show that GalNAc-T relocalisation is rapidly reversible and suggests that ERK8 provides a continual brake for GalNAc-T relocation at the Golgi.

## ERK8 localises at the Golgi and is displaced upon growth factor stimulation

GalNAc-Ts are thought to regularly cycle between the ER and the Golgi apparatus (*Rhee et al., 2005*). Thus, relocation of these glycosylation enzymes to the ER upon ERK8 depletion could result either from an enhanced export from the Golgi or an inhibition of exit from the ER. To address this, we first analysed the subcellular localisation of ERK8 protein by immunofluorescence and observed a predominantly cytosolic pattern in wild-type HeLa cells. However, prolonged permeabilisation (2 hr) clearly revealed positive Golgi staining, suggesting that a fraction of ERK8 is associated with this organelle (*Figure 4A*). This is consistent with the perinuclear pattern reported in A-431 cells by the Protein Atlas project (*Uhlen et al., 2010*).

Next, HeLa cells were stimulated with 50 ng/µl of PDGF, resulting in an increase in HPL staining intensity between 30 min and 2 hr and a decrease in ERK8 at the Golgi apparatus (*Figure 4B*). Using the Pearson's Correlation coefficient of ERK8 and TGN46 staining, we found a 60% decrease after 2 hr, suggesting that ERK8 is displaced from the Golgi after cell stimulation (*Figure 4C*).

SRC is a key signal transducer between PDGF and GalNAc-T traffic. A mutant, inactive form of SRC (Src-8A7F) can be re-activated using imidazole (*Qiao et al., 2006*). Using a HeLa cell line stably expressing Src-8A7F-mCherry, we observed a gradual decrease in ERK8 at the Golgi (*Figure 4—figure supplement 1A,B*), whereas no change was observed with a catalytically defective SRC mutant (Src 6N7F) (*Figure 4—figure supplement 1C*). This suggests that SRC activity regulates ERK8 localisation at the Golgi.

Overall, our data indicate that ERK8 is dynamically localised at the Golgi apparatus where it likely controls GalNAc-T export.

## ERK8 regulates COPI-dependent GalNAc-Ts traffic

To test if the relocation of GalNAc-Ts in ERK8-depleted cells is dependent on COPI, we first expressed the dominant-negative mutant of Arf1(Q71L), which is unable to hydrolyse bound GTP (*Dascher and Balch, 1994*) and found significant rescue of Tn levels in contrast with cells expressing wild-type Arf1 (*Figure 5—figure supplement 1A*). ERK8-depleted cells were also treated with 50 nM of the GBF1 inhibitor Golgicide (*Saenz et al., 2009*), which also provided significant rescue (*Figure 5—figure supplement 1B*).

Consistent with these results, combined knockdown of ERK8 and GBF1 almost completely reversed high HPL staining, further indicating that GBF1 is required for GalNAc-T relocation from the Golgi to the ER (*Figure 5A*). Co-knockdown of ERK8 with Arf1, -3, -4 or -5 also reduced Tn levels by about 60% (*Figure 5B*). Combined knockdowns of Arfs further increased the rescue, suggesting functional redundancy amongst the Arf proteins. By contrast, co-knockdown with Arf6, which does not regulate COPI, did not affect Tn levels (*Figure 5B*). These reductions in HPL levels were not due to reduced knockdown efficiencies of ERK8, as similar effects were observed with increasing amounts

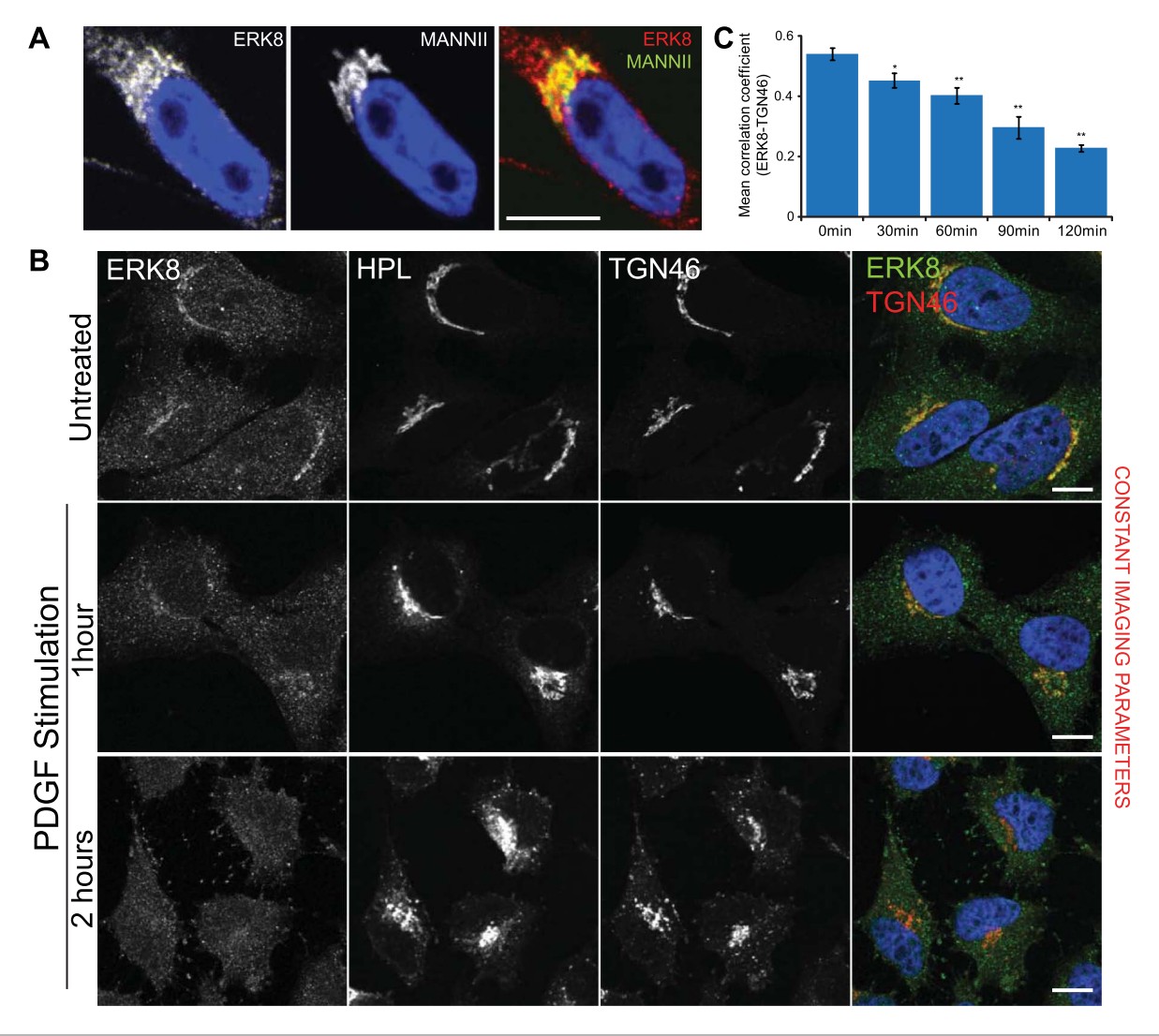

Figure 4. ERK8 is dynamically localised at the Golgi. (A) High magnification of HeLa MannII-GFP expressing cells stained for endogenous ERK8 following cytosol extraction. (B) Cytosol-depleted cells treated with platelet-derived growth factor (PDGF; 50 ng/ml) for the indicated times and stained for ERK8, Tn (*Helix pomatia* lectin, HPL) and the Golgi marker, TGN46. Scale bar: 10 µm. (C) Pearson's correlation coefficient between ERK8 and Golgi marker TGN46 in cells treated with PDGF for the indicated times. Values on graphs indicate the mean ± SEM. **p<0.0001, *p<0.05 by two-tailed unpaired *t* test, relative to vehicle treated cells.

The following figure supplements are available for figure 4:

**Figure supplement 1**. ERK8 is dynamically localised when SRC is increasingly activated at the Golgi.

of non-targeting (NT) siRNA added to the transfection mix (*Figure 5—figure supplement 1C*). The efficiency and specificity was also verified by assaying protein expression of each gene (*Figure 5— figure supplement 1D*). Collectively, these data indicate that the COPI trafficking machinery is essential for the ER relocation of GalNAc-Ts.

A key activation step for the COPI coatomer is the exchange of GDP for GTP on Arf1 (*Antonny et al., 2005*; *Beck et al., 2009*; *Szul and Sztul, 2011*). Therefore, we assessed Arf1-GTP loading after ERK8 inhibition using Ro-31-8220 and found activation of Arf1 as early as 15 min and sustained for over 2 hr (*Figure 5C*).

To evaluate the effect of ERK8 depletion on COPI, we next stained cells depleted by siRNA for the Golgi marker GM130 and the COPI subunit beta-COP (COPB). As previously observed with

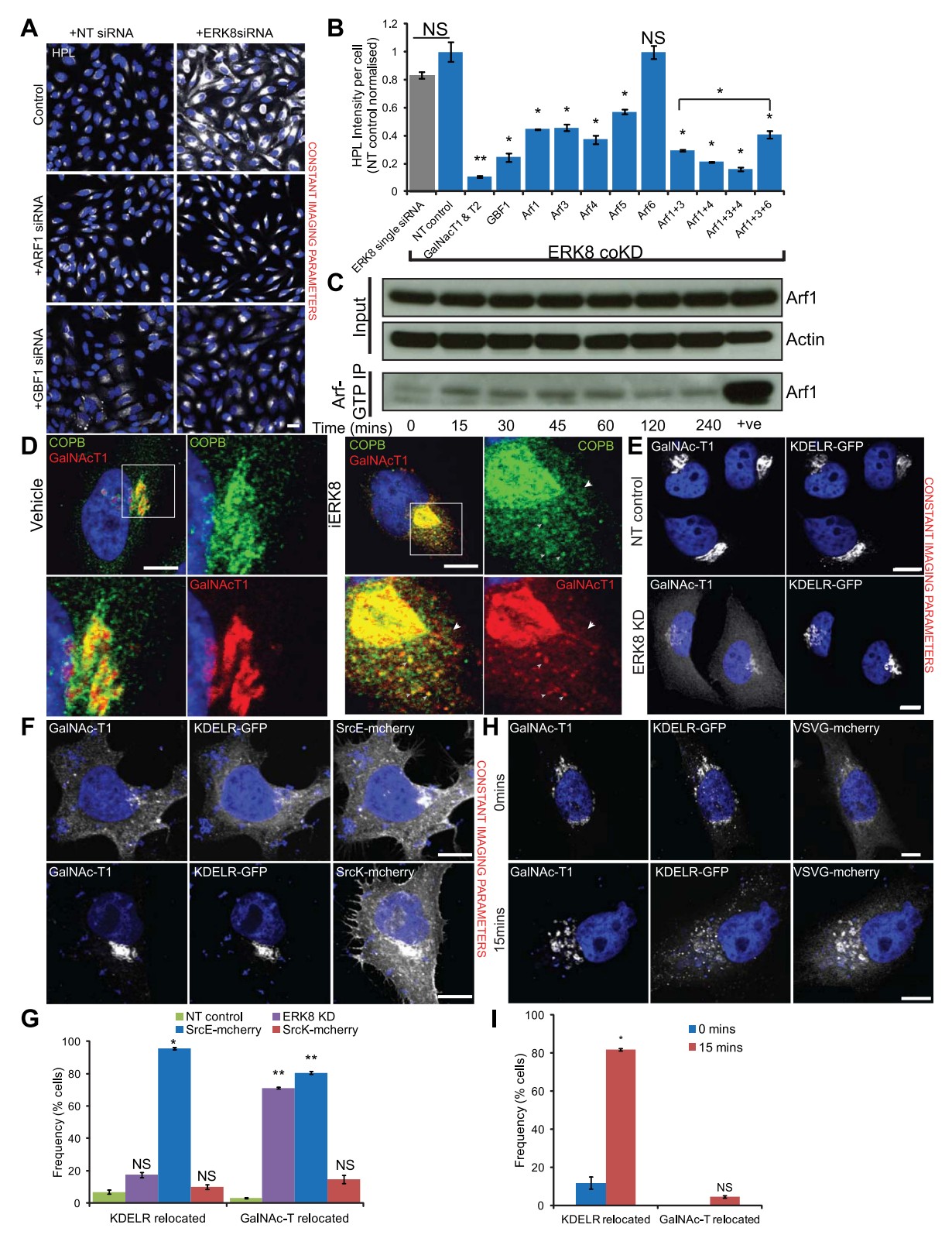

**Figure 5**. ERK8 regulates COPI-dependent GalNAc-T traffic. (**A**) Co-knockdown of ERK8 with Arf1 or GTP exchange factor, GBF1, and staining with *Helix pomatia* lectin (HPL). NT, non-targeting. Scale bar: 30 μm. (**B**) Quantification of Tn levels upon ERK8 co-knockdown with Arf proteins and GBF1. Grey bar indicates knockdown of ERK8 only. Blue bars indicate co-knockdowns. (**C**) SDS-PAGE analysis of total Arf and Arf1-GTP in cells treated with 5 μM

*Figure 5. Continued on next page*

*Figure 5. Continued*

Ro-31-8220 (iERK8). (**D**) Co-staining of Beta-COP (COPB) and GalNAc-T1 in cells treated with 5 µM iERK8 for 15 min. Transient tubular structures emanating from the Golgi appear stained for GalNAc-T1 and beta-COP (arrowhead, second panel). Scale bar: 10 µm. (**E**) Effect of ERK8 depletion on GalNAc-T1 and KDEL receptor (KDEL-R) subcellular location. (**F**) Effect of expression of active SRC (SrcE-mcherry, containing the E378G mutation) or inactive SRC (SrcK-mcherry, containing the K295M mutation) on both proteins. Scale bar: 10 µm. (**G**) Visual scoring of KDEL-R and GalNAc-T1 redistribution from the Golgi in cells subjected to various treatment conditions. Cells were counted in each condition from three independent experiments (NT control: 83 cells; ERK8 KD: 86; SrcE-mcherry: 42; SrcK-mcherry: 32). (**H**) Temperature-sensitive vesicular stomatitis virus G glycoprotein (VSVG-mcherry) traffic to the Golgi induced by shift from restrictive to permissive temperature for 15 min in KDEL-R expressing cells stained for GalNac-T1. Scale bar: 10 µm. (**I**) Visual scoring of KDEL-R and GalNAc-T1 relocation in VSVG expressing cells at 0 and 15 min after temperature shift. Cells were counted in each condition from three independent experiments (0 min, 44 cells; 15 min, 63 cells). Values on graphs indicate the mean ± SEM. \*\*p<0.0001, \*p<0.05 by two-tailed unpaired *t* test, relative to NT siRNA-treated (**B** and **G**) cells and cells at 0-min timepoint (**I**).

The following figure supplements are available for figure 5:

**Figure supplement 1**. ERK8 regulated GalNAc-T traffic depends on the activity of COPI regulators.

MannII-GFP, GM130 staining revealed significant Golgi fragmentation after ERK8 depletion (*Figure 5—figure supplement 1E*). Interestingly, COPB staining was significantly more affected and found on small structures in the cytoplasm, suggesting enrichment on transport intermediates (*Figure 5—figure supplement 1E*). Using a granularity measurement algorithm, we found a nearly 4-fold increase in distribution for COPB but only slightly more than a twofold increase for GM130 (*Figure 5—figure supplement 1F*).

When treated with the ERK8 inhibitor Ro-31-8220, cells displayed a significant redistribution of COPI coatomer staining as early as 5 min after treatment (*Figure 5—figure supplement 1G,H*). Furthermore, in cells inhibited by Ro-31-8220 for 15 min, numerous COPI-positive vesicular structures were clearly co-stained with GalNAc-T1 antibodies (*Figure 5D*). In several instances, we could detect tubular structures emanating from the Golgi apparatus that stained positively for GalNAc-T1 as well as COPB, although not as homogenously along their length as for seen for GalNAc-T1 (*Figure 5D*).

A well-described cargo of COPI carriers in Golgi-to-ER retrograde traffic is the KDEL receptor (KDEL-R) (*Orci et al., 1997*). KDEL-R trafficking can be induced by a wave of cargo or by SRC activation (*Bard et al., 2003*; *Pulvirenti et al., 2008*). However, we found that ERK8 depletion did not detectably affect KDEL-R distribution in cells where GalNAc-T relocation was extensive (*Figure 5E*, *Figure 5—figure supplement 1I*). By contrast, expression of an active form of SRC (E378G; SrcE) similarly relocated GalNAc-T1 to the ER, as compared with inactive SRC (K295M; SrcK) but it also strongly affected KDEL-R-GFP distribution (*Figure 5F*). Visual scoring of the localisation of KDEL-R and GalNAc-T1 revealed than more than 80% of the cells display relocation for both proteins in SrcE expressing cells. By contrast, while more than 70% of ERK8 depleted cells exhibit clear GalNAc-T redistribution from the Golgi, less than 20% show KDEL-R relocation (*Figure 5G*). These results indicate that GalNAc-Ts and KDEL-R trafficking are differentially regulated.

Consistent with this observation, when a wave of the temperature-sensitive mutant of the vesicular stomatitis virus G glycoprotein (VSVG) was induced by temperature shift, KDEL-R was relocated from the Golgi to the ER as previously reported (*Figure 5H*; *Pulvirenti et al., 2008*), whereas GalNAc-Ts were not affected (*Figure 5H,I*).

Altogether, these results suggest that SRC stimulates both GalNAc-Ts and KDEL-R COPI-dependent retrograde traffic whereas ERK8 inhibits specifically the formation of transport intermediates containing GalNAc-Ts.

## ERK8 regulates cell migratory ability through control of O-glycosylation

O-glycosylation in the ER stimulates cell adhesion and cell migration and tends to induce a spindle-shaped morphology (*Gill et al., 2013*). Interestingly, this morphology was also apparent in ERK8-depleted HeLa cells under phase contrast microscopy (*Figure 6A*) and after staining for the actin and tubulin cytoskeletons (*Figure 6B*).

When tested on fibronectin-coated plates in a scratch-wound healing assay, ERK8-depleted HeLa cells migrated about twofold faster into the denuded area compared with NT siRNA-treated cells (*Figure 6C*). This faster rate was constant over 7 hr (*Figure 6D*), indicating that the faster wound closure is caused by faster cell migration and not enhanced reactivity to the initial wound. ERK8 knockdown

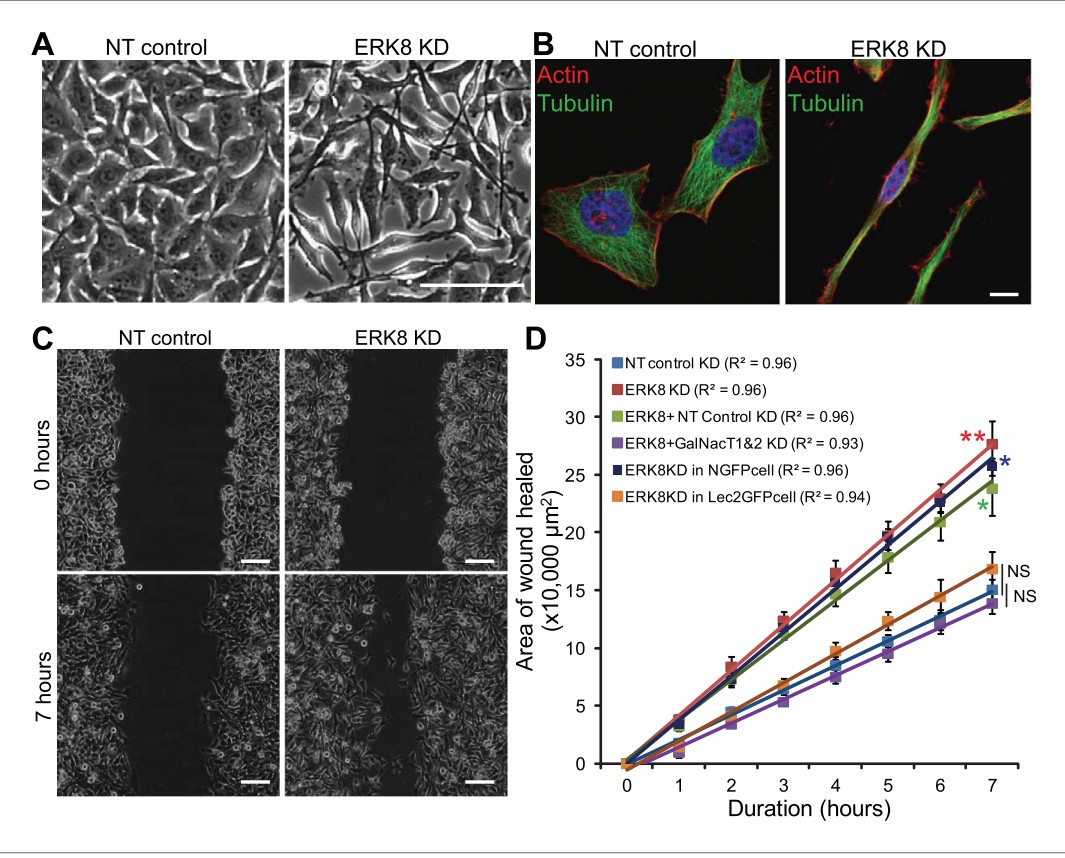

**Figure 6**. ERK8 regulates cell migration through ER O-glycosylation. (**A**) Phase contrast images and (**B**) actin and tubulin staining of non-targeting (NT) siRNA-treated and ERK8-depleted cells. Scale bars: 100 µm in (**A**) and 10 µm in (**B**). (**C**) Migration assay using scratch wound of cellular monolayer in NT siRNA-treated and ERK8-depleted cells. Scale bar: 100 µm. (**D**) Rate of wound closure (area) measured over 7 hr (n = 4 experiments for each condition). Values on graphs indicate mean ± SEM. **p<0.001, *p<0.05 by two-tailed unpaired *t* test. Red asterisks indicate *t* test between NT siRNA-treated and ERK8-depleted cells. Green asterisk indicates t-test between cells co-knockdown with ERK8 and GalNAc-T1 and -T2 (ERK8+GalNAc-T1 & -T2 KD) and cells co-knockdown with ERK8 and NT siRNA (ERK8+NT control KD). Blue asterisk indicates *t* test between ERK8 knockdown in NGFP-expressing and ER-localised GalNAc-T inhibitor Lec2GFP cells. NS, not significant (black vertical lines).

The following figure supplements are available for figure 6:

**Figure supplement 1**. ERK8 inhibits cell motility by controlling Tn expression on cell-surface O-glycoproteins.

also led to a dramatically higher cell surface Tn staining, with numerous Tn-positive protrusion structures (*Figure 6—figure supplement 1A*). These Tn-bearing glycoproteins are likely to promote increased cell adhesion, as shown previously (*Gill et al., 2013*).

However, ERK8 has also been implicated in various other cellular processes. To verify that the increased cell motility was due to enhanced O-glycosylation, the scratch-wound healing assay was repeated in ERK8 and GalNAc-T1 and -T2 knocked down cells (ERK8+GalNAcT1&2 KD). We found that these cells migrated significantly slower than ERK8 and ERK8+NT knockdown cells and were similar to NT control cells (*Figure 6D*, *Figure 6—figure supplement 1B*). To further confirm the importance of ER O-glycosylation, we used the ER-localised GalNAc-T inhibitor, Lec2GFP. Cell migration rates induced by ERK8 depletion in Lec2GFP cells were significantly reduced compared with cells expressing only GFP (NGFP cells) (*Figure 6D*, *Figure 6—figure supplement 1C*) and, again, were rather similar to NT control cell migration. It is important to note that the Lec2GFP construct itself did not significantly slow cell migration in the absence of ERK8 depletion.

Thus, collectively, our results indicate that ERK8 is a negative regulator of cell migration through inhibition of protein O-glycosylation in the ER.

## ERK8 expression is frequently downregulated in breast and lung carcinoma

An initial goal of this study was to elucidate mechanisms for the increase in Tn levels frequently observed in tumours. Interestingly, ERK8 protein levels are reported to be fairly constant in normal breast tissue and benign tumours but drop significantly in malignant tumours, with a loss measured in approximately 50%, 80% and 100% of grade 1, 2 and 3 tumours, respectively (*Henrich et al., 2003*). Henrich et al. proposed that ERK8 stimulates the degradation of Oestrogen Receptor-alpha, suggesting a possible selective advantage for the loss of this MAPK (*Henrich et al., 2003*). Interestingly, this trend is also consistent with the increased frequency and intensity of Tn staining (*Gill et al., 2013*).

To further examine this question, both ERK8 and Tn (stained with VVL) were co-labelled in a panel of 39 frozen tissue arrays comprising 5 normal and 34 invasive ductal breast carcinoma (*Figure 7—figure supplement 1A–C*). Quantification of ERK8 levels of each tissue core was performed by measuring the area above a fixed threshold, normalised to the total area of the core represented by nuclei staining (DAPI) (*Figure 7—figure supplement 1D*). Although levels varied considerably, more than half of the carcinoma cores (18/34) showed at least 50% lower expression of ERK8 (*Figure 7A*). Tn levels also varied significantly but, in the large majority of samples, they were significantly higher in tumour samples as compared with normal cores (*Figure 7B*, *Figure 7—figure supplement 1C,D*). In most cores, ERK8 and Tn levels appeared to show opposing trends, suggesting that the loss of ERK8 could partially drive high Tn expression. However, there was no clear correlation between the levels of both antigens (*Figure 7—figure supplement 1E*).

High Tn has also been reported in other tumour types, where oestrogen regulation is not thought to be critical. As ERK8 was previously found to be highly expressed in the lung (*Abe et al., 2002*), we set out to explore the link between ERK8 and Tn levels in lung cancer with 23 lung biopsies containing 2 normal lung tissues, 14 squamous cell carcinomas, 6 adenocarcinomas and 1 small cell carcinoma. We found that ERK8 was clearly detectable in normal tissues but that the levels appeared markedly lower in all lung carcinomas (*Figure 7—figure supplement 1H*), with an average 80% loss of expression, and a range of 40–90% (*Figure 7C*). No specific trend in regard to the cancer type was noticed. Next, we quantified Tn levels (*Figure 7—figure supplement 1F*) and found a significantly higher expression of Tn in a majority of the samples, with a more than fourfold average higher expression (*Figure 7D*, *Figure 7—figure supplement 1I*) and a range of 2–10-fold; 18 cores (86%) displayed higher than normal levels of Tn (*Figure 7D*).

We next explored if high Tn levels were linked to ER O-glycosylation, as in the case of breast carcinomas and indeed found that Tn staining co-localised extensively with the ER marker calnexin in the carcinoma samples. By contrast, Tn staining in the normal lung tissue was clearly more punctuate and reminiscent of a Golgi localisation (*Figure 7E*).

Thus, in most lung carcinoma tumours, ERK8 expression is lower while Tn levels are higher as compared with normal lung tissue. Furthermore, in samples with heterogeneous staining for Tn, we observed a reverse correlation between ERK8 and Tn staining (*Figure 7F,G*). However, at the whole core level, the staining intensities of the two levels were not inversely correlated from sample to sample (*Figure 7—figure supplement 1J*).

The lack of a direct correlation indicates that, in human lung and breast tumours, ERK8 levels do not strictly control the levels of Tn. Given the number of regulators that have been already identified in this and previous studies, this is not surprising. Notwithstanding, our analysis indicates that ERK8 is frequently downregulated in lung and breast carcinomas, which probably facilitates the relocation of GalNac-Ts to the ER when other biochemical or genetic perturbations, such as SRC activation, are engaged.

## Discussion

Cellular levels of the Tn antigen vary dramatically in cancer cells, suggesting that O-glycosylation initiation and/or elongation is highly regulated. Indeed, our screen results reveal that several signalling molecules exert significant control over O-glycosylation initiation. In recent years, we have reported that this initiation step can be regulated through trafficking of the GalNAc-Ts between the ER and the Golgi (*Gill et al., 2010*) and that marked ER localisation explains the high Tn phenotype in at least 60% of human breast tumours (*Gill et al., 2013*). High Tn levels can also arise from a loss or inhibition of the elongation of the O-GalNAc, a mechanism that has also been proposed to underlie the cancer phenotype (*Ju et al., 2011*).

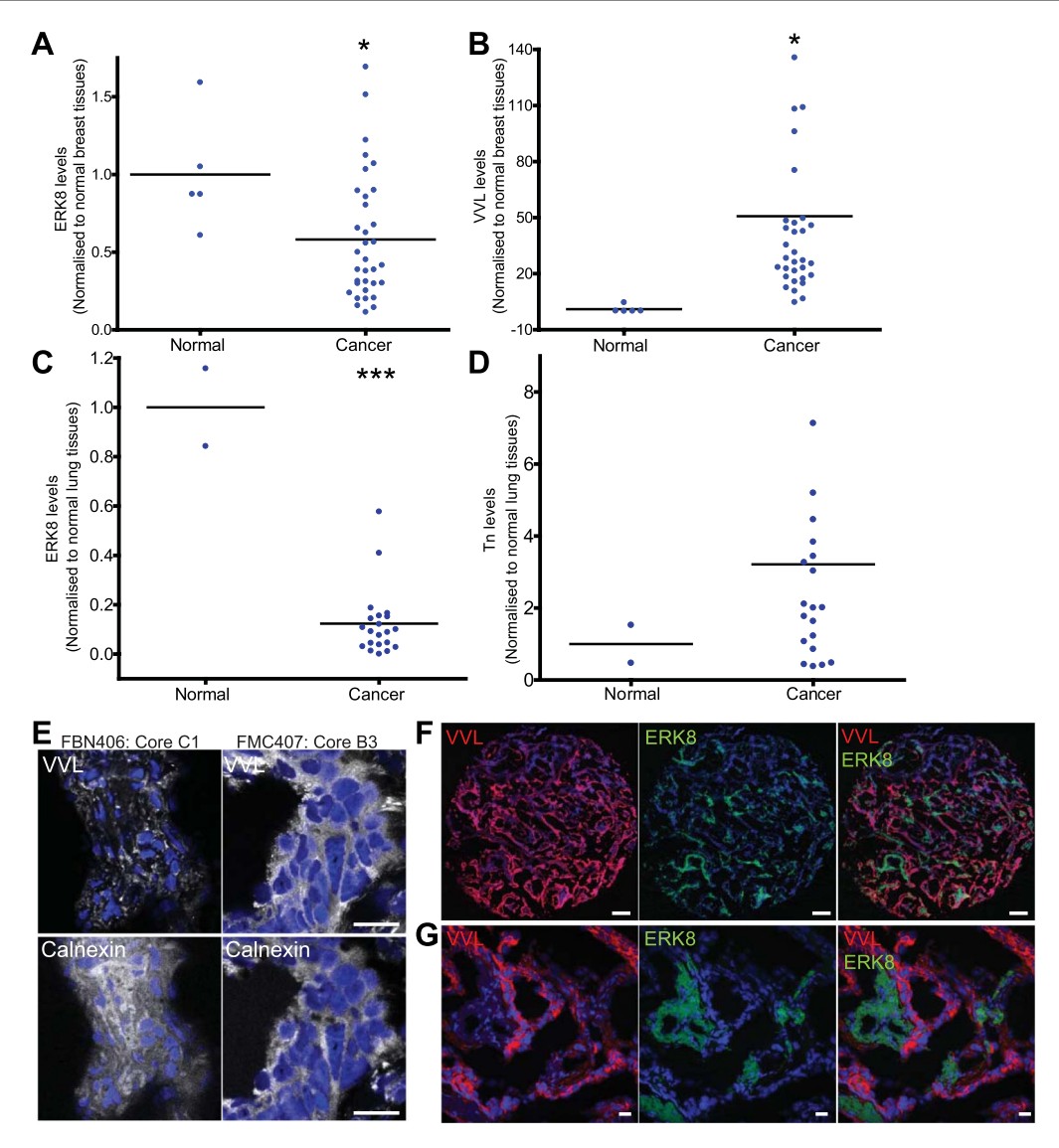

**Figure 7**. ERK8 is downregulated in human breast and lung carcinoma. (**A**) Quantification of ERK8 staining in human breast biopsies. Each point represents the staining of one tissue core normalised to the average staining of the normal tissue cores. (**B**) Quantification of Tn (*Vicia Villosa* Lectin; VVL) staining in human breast biopsies. (**C**) Quantification of ERK8 staining in human lung biopsies. (**D**) Quantification of Tn staining in human lung biopsies. *p<0.05, **p<0.01, ***p<0.0001 by two-tailed unpaired *t* test. (**E**) Co-staining VVL and ER marker Calnexin revealed extensive ER co-localisation of Tn in lung carcinoma (FMC407: Core B3), whereas Tn appeared as punctuate structures in the normal lung (FBN406: Core C1). Scale bar: 20 µm. (**F**) ERK8 and Tn staining in a lung adenocarcinoma core (FMC407: Core B8). Scale bar: 200 µm. (**G**) Close-up image of the core shown in (**F**). Scale bar: 20 µm.

The following figure supplements are available for figure 7:

**Figure supplement 1**. ERK8 levels are frequently reduced in human lung carcinoma.

In secondary screens, it appeared that most, if not all, signalling proteins affecting Tn levels regulate the subcellular localisation of GalNAc-Ts and not the elongation process. This obviously does not preclude the main elongation enzyme, C1GALT or its specific chaperone, COSMC (alias *C1GALT1C1*) from being regulated in some conditions, but a significant inhibition of the activity of these proteins was not observed in our screen conditions. In contrast, the subcellular localisation of GalNAc-Ts appears to be a nodal point of control in a complex signalling network. Indeed, at least 12 independent

negative regulators were identified and at least as many positive regulators, including the SRC family tyrosine kinases, are likely to be involved. This regulatory complexity suggests that perhaps signals of different origins are being integrated at the level of GalNAc-Ts traffic.

ERK8 is one of the most potent regulators we identified. Multiple pieces of evidence indicate that it acts at the level of the Golgi by inhibiting the formation of GalNAc-T-containing COPI vesicles. Based on their genetic interaction profile, the other negative regulators appear to act at the same level as ERK8. Consistently, two of these proteins, PKMYT1 and MAP4K2, are also reported to localize at the Golgi (*Ren et al., 1996*; *Liu et al., 1997*), and several other regulators interact with Golgi-localised proteins. Together, this suggest that the incorporation of GalNAc-Ts in COPI vesicles is the key point of regulation of this potential regulatory network.

This regulation point also reveals the existence of at least two different types of COPI-dependent Golgi-to-ER retrograde traffic carriers: one type of transport, GalNAc-Ts, is activated by growth factor stimulation and cancerous transformation and is repressed by ERK8; the other, KDEL-R, is activated by cargo protein traffic and is independent of ERK8. However, both routes appear to be under the control of the SRC kinase family. A key step in understanding these differences will be to identify the relevant phosphorylation substrates for both SRC and ERK8.

The trafficking of GalNAc-Ts to the ER results in the glycosylation of multiple different substrates, as indicated by the metabolic labelling results. The precise identity of these substrates, as well as the functional effects of their glycosylation, remains to be established. Notwithstanding, the outcome of the relocation at the cellular level is clearly a significant stimulator of cell migration. Indeed, the stimulatory effect of ERK8 depletion is dependent on ER-localised O-glycosylation. These results are also consistent with our previous data based on the expression of an exogenous, ER-targeted form of GalNAc-T2 (*Gill et al., 2013*). Thus, an interesting hypothesis is that the intensity of packaging of GalNAc-Ts into Golgi-derived COPI-coated vesicles could be a signalling integration point that sets the 'motility potential' of cells.

In breast and lung cancer cells, this set-point appears constitutively high, as the relocation of GalNAc-Ts is extensive and frequent. The promotion of cell motility associated with ER-localised O-glycosylation appeared to favour the formation of lung metastases in a tail-vein injection-based assay (*Gill et al., 2013*). Thus, how GalNAc-T relocation is stimulated in cancer cells has probably important medical implications. Our analyses suggest that multiple mechanisms are possible, including a decrease in ERK8 protein levels. However, the actual level of decrease required to stimulate relocation is not known and is anyway probably dependent on other cellular parameters. Additionally, Tn levels are not likely to depend only on the intracellular distribution of GalNAc-Ts. For instance, normal tissues with high levels of mucin expression, such as stomach, colon or kidney, tend to have higher endogenous Tn levels without clear evidence for relocation. This complexity probably contributes to the lack of direct correlation between ERK8 and Tn levels.

In addition to promoting O-glycosylation, ERK8 depletion could also have other beneficial advantages for cancer cells. Indeed, ERK8 has been implicated in multiple, apparently unrelated, molecular processes, such as maintenance of genomic integrity (*Groehler and Lannigan, 2010*), regulation of telomerase activity (*Cerone et al., 2011*), autophagy (*Colecchia et al., 2012*) and inhibition of nuclear receptor activity (*Henrich et al., 2003*; *Saelzler et al., 2006*; *Rossi et al., 2011*). Recently, the *Drosophila* homolog Erk7 and human ERK8 were also shown to participate in the regulation of protein secretion during starvation through disassembly of ER exit sites (*Zacharogianni et al., 2011*). Whether these different processes are somewhat linked through ERK8 or whether ERK8 is simply moonlighting in different, independent functions constitutes an interesting challenge for the future.

In sum, our results suggest that initiation of O-glycosylation in the ER is under an elaborate regulatory control system of which ERK8 is a key player. This regulation sets the level of cellular motility and is frequently perturbed in cancer cells of breast and lung origins.

## Materials and methods

### Cloning and cell culture

HeLa MannII-GFP was from Vivek Malhotra's laboratory (CRG, Barcelona) and maintained in DMEM with 10% fetal bovine serum (FBS). HeLa cells that were knockout of COSMC was obtained from U Mendel and H Clausen (University of Copenhagen, Denmark). HeLa ER-Lec2-GFP and KDEL-R-GFP

cell lines were generated by lentiviral infection of HeLa wild-type cells with ER-2Lec-GFP (*Gill et al., 2013*) and KDEL-R-GFP lentivirus and subsequently, FACS sorted to enrich for GFP-expressing cells. HEK293T cells were grown in DMEM supplemented with 15% FBS. All cells were grown at 37°C in a 10% $CO_2$ incubator. Human *MAPK15/ ERK8* (NM_139021) was amplified from cDNA purchased from Origene (#RG216589; Rockville, MD) by PCR and cloned into entry vector pDONR221 (Invitrogen, Life Technologies Corporation, Carlsbad, CA). The catalytically inactive ERK8 mutant construct was generated by introducing T175A and Y177F mutations using the QuikChange Site-Directed Mutagenesis Kit (Stratagene, Amsterdam, The Netherlands). The entry vectors were subsequently cloned into pcDNA6.2-Nmcherry-DEST, a gateway compatible destination vector constructed by the replacement of the GFP tag with mCherry on pcDNA6.2-NGFP-DEST (Invitrogen). Human ARF1 (NM-001024228)-GFP expression clones were described previously (*Gill et al., 2010*). miR-ERK8 and miR-GFP vector were generated from BLOCK- iT Pol II miR RNAi expression vector kits from Invitrogen. All constructs were verified by sequencing and restriction enzyme digests before use.

## Antibodies and reagents

*Helix pomatia* Lectin (HPL) conjugated with 647 nm fluorophore (#L32454), Alexa Fluor secondary antibodies, and Hoechst 33342 (#H3570) were purchased from Invitrogen. siRNAs were obtained from Dharmacon (Thermo Fisher Scientific, Wilmington, DE). OptiMEM was purchased from Invitrogen, and Hiperfect transfection reagent was purchased from Qiagen (Valencia, CA). Anti-GalNAc-T1, GalNAc-T2 and Tn hybridomas for immunofluorescence staining were a gift from U Mendel and H Clausen (University of Copenhagen, Denmark). Anti-COPI coatomer (targeting native coatomer) was a gift from FT Wieland (University of Heidelberg, Germany). Anti-GRASP55 was a gift from Vivek Malhotra's laboratory (CRG, Barcelona). Anti-GalNAc-T1 (#sc-68491) for western blotting was purchased from Santa Cruz Biotechnology (Dallas, TX). Anti-ERK8 antibody (#HPA002704) was purchased from Sigma–Aldrich (St Louis, MO). Anti-beta COP antibody (#ab2899), anti-Giantin (#ab24586), anti-C2GNT1 (#ab38858) and anti-COSMC (#ab93483) were from Abcam (Cambridge, MA). Fluorescein-labelled *Vicia Villosa* Lectin (VVL) (#FL-1231) and Rhodamine-labelled Peanut Agglutinin PNA (#RL-1072) were from Vector laboratories Inc. (Burlingame, CA). Ro-31-8220 (#557521) and α-Amanitin (#129741) was from Merck (Rahway, NJ). Golgicide A (GCA) (#G0923) was from Sigma–Aldrich. *ERK8*-specific siRNA sequences (dERK8-1: 5'-GUAGUGGACCCUCGCAUUG-3', dERK8-2: 5'-AGAACGACAGGGACAUUUA-3', dERK8-3: 5'-GGAGAUACCUACUCAGGCG-3' and dERK8-4: 5'-CCUAUGGCAUUGUGUGGAA-3') were purchased from Thermo-fisher.

## siRNA screening

siRNA transfection, immunofluorescence staining and imaging procedures were described in detail previously (*Chia et al., 2012*).

## Automated image acquisition and quantification

During automated image acquisition, four sites per well were acquired sequentially with a 20 × Plan Apo 0.75 NA objective on a laser scanning confocal high-throughput microscope (ImageXpress Ultra, Molecular Devices, Sunnyvale, CA). Image analysis was performed using MetaXpress software (version 3.1.0.89). For each well, total HPL staining intensity and nuclei number was quantified using Transfluor HT application module in the software. Briefly, masking for both Cy5 (HPL) and DAPI (Nuclei) channels was generated by setting the mask dimensions and cut-off intensity above the background for each of the two channels (*Figure 1A*) and intensities were quantified in the area covered by the masking. Hundreds of cells were quantified and the averages per well were calculated. To compare HPL intensities between wells, HPL intensity per cell of each well was obtained by normalising the total HPL intensity ('Integrated Granule Intensity' of Cy5 channel) with nuclei number of the well. The same procedures were performed for the quantification of PNA and GalNAc-T1 staining in the secondary screens. Statistical significance was measured using a paired t test assuming a two-tailed Gaussian distribution.

## Drug treatments

For ERK8 inhibitor Ro-31-8220 treatment, HeLa MannII-GFP cells were seeded into imaging plates overnight, treated with 5 μM Ro-31-8220 in DMEM with 10% FBS for various durations and then fixed with 4% paraformaldehyde-4% sucrose in phosphate-buffered saline (PBS) followed by subsequent staining. For Golgicide A treatment, cells were treated with 50 nM of Golgicide A and fixed at different time points.

## High-resolution fluorescence microscopy

Cells were seeded onto glass coverslips in 24-well dishes (Nunc, Denmark). After the respective treatments, cells were fixed with 4% paraformaldehyde-4% sucrose in D-PBS, permeabilised with 0.2% Triton-X for 10 min and stained with the appropriate markers using the same procedure performed in the primary siRNA screen. To effectively observe ERK8 localisation at the Golgi, the cells were permeabilised with 0.2% Triton-X for 2 hr and stained with anti-ERK8 antibody diluted in 2% FBS in D-PBS overnight. For beta-COP (COPB) and COPI coatomer staining, cells were permeabilised with 0.05% NP40 for 5 min twice, washed with D-PBS twice for 5 min, blocked with 2% bovine serum albumin (BSA) for 1 hr at room temperature, and then stained with primary antibody diluted in 2% FBS in D-PBS overnight. Cells were mounted onto glass slides using FluorSave (Merck) and imaged at room temperature using an inverted confocal microscope (IX81; Olympus Optical Co. Ltd, Tokyo, Japan) coupled with a CCD camera (model FVII) either with a 60 × objective (U Plan Super Apochromatic [UPLSAPO]; NA 1.35) or 100 × objective (UPLSAPO; NA 1.40) under Immersol oil. Images were acquired at 100x magnification and processed using Olympus FV10-ASW software.

## O-glycosylation reporter analysis

### GalNaz metabolic labelling

HeLa cells were treated with siRNA for 3 days before metabolically labelling with 20 μM GalNAz for 6 hr. Ro-31-8220 was added to cells during GalNAz metabolic labelling for 4 hr treatment before harvesting. Cells were washed twice using ice-cold D-PBS before scraping in D-PBS. Cells were centrifuged at 300×$g$ for 5 min at 4°C and were lysed with ice-cold lysis buffer (50 mM Tris [pH 8.0, 4°C], 200 mM NaCl, 0.5% NP-40 alternative and complete protease inhibitor [Roche Applied Science, Mannheim, Germany]) for 30 min with gradual agitation before clarification of samples by centrifugation at 10000×$g$ for 10 min at 4°C. Clarified lysate protein concentrations were determined using Bradford reagent (Bio-Rad Laboratories, Hercules, CA) before sample normalisation. Samples were diluted in lysis buffer with 4 × SDS loading buffer and boiled at 95°C for 2 min. They were then resolved by SDS-PAGE electrophoresis using bis-tris NuPage gels as per the manufacturer's instructions (Invitrogen) and transferred to PVDF membranes. Membranes were then blocked using 3% BSA dissolved in Tris-buffered saline with tween (TBST: 50 mM Tris [pH 8.0, 4°C], 150 mM NaCl, and 0.1% Tween 20) for 2 hr at room temperature. Membranes were washed to remove traces of BSA before incubation with antibodies, as per the manufacturer's instructions. Membranes were washed five times with TBST before incubation with secondary HRP-conjugated antibodies (GE Healthcare). Membranes were further washed five times with TBST before ECL exposure.

### ER-specific GalNAc-T activity reporter assay

The experiment was performed as described previously (*Gill et al., 2010*). Briefly, HEK293T cells were seeded into 10-cm petri dishes 24 hr before transfection and were then transfected with the ER-trapped mucin construct using the calcium phosphate method. The growth media was then replaced the following day and the cells were further incubated for 24–48 hr before ERK8 inhibitor treatment for 4 hr. Cells were harvested and lysed as described above. All subsequent steps were performed either on ice or at 4°C. Immunoprecipitation (IP) samples were incubated with 1–2 μg of HPL-conjugated agarose (Sigma–Aldrich) overnight at 4°C. The next day, IP samples were washed five times with 1 ml of IP wash buffer (50 mM Tris [pH 8.0, 4°C], 100 mM NaCl, 0.5% NP-40 alternative, 1 mM DTT, and complete protease inhibitors [Roche]). Samples were diluted in IP wash buffer and 4 × SDS loading buffer before boiling at 95°C for 2 min. Samples were then resolved by SDS-PAGE electrophoresis as described above.

## VSVG pulse-chase assay

HeLa KDELR-R-GFP expressing cells were transfected to express the temperature-sensitive mutant of vesicular stomatitis virus G glycoprotein (VSVG-tsO45) tagged with mcherry at the C-terminus. For the VSVG pulse-chase experiment, cells were transferred to 40°C for 16 hr and then shifted to 32°C for various durations in the presence of 100 μg/ml of cycloheximide before fixation. Cells were imaged at 100x magnification and quantified by eye for KDEL-R and GalNAc-T subcellular localisation.

## Wound healing assay

Cells were seeded onto fibronectin-coated 35-mm plastic tissue culture dishes (Ibidi GmbH) and grown to confluence (16–24 hr). A wound was generated using a micropipette tip before washing to remove

cell debris. Live phase contrast imaging was performed at 37°C using a Zeiss Axiovert microscope (model 200M; Zeiss Microimaging; Thornwood, NY) equipped with a CCD camera (AxioCam HRc) and a 20 × objective (LD Plan-NEOFLUAR; 20 ×; N.A. 0.4). Frames were acquired at 5-min intervals. Areas of wound invasion were calculated using ImageJ (National Institutes of Health, Bethesda, MD).

## Human tumour microarray imaging and quantification

Frozen human tumour microarrays FBN406a and FMC407 were purchased from US Biomax, Inc. (Rockville, MD). Briefly, the slides were dried and fixed in chilled in a 1:1 acetone:methanol solution for 10 min at room temperature. The slides were then washed three times with TBST and blocked with 10% goat serum-PBS for 30 min. Subsequent staining with ERK8 antibody (0.9 µg/ml), VVL-biotin (4 µg/ml) and Hoescht (1:10,000) was performed overnight before staining with anti-rabbit Alexa Fluor 488 (1:1000) and Streptavidin-Alexa 594 (1:400) secondary antibodies for 30 min. Slides were counterstained with DAPI and then mounted (Vectashield). The arrays were first automatically imaged (using constant acquisition parameters) using a 10 ×objective (LD Plan-NEOFLUAR; 10 ×; N.A. 0.4) on a motorised stage coupled to a Zeiss inverted confocal microscope equipped with a CCD camera (AxioCam HRc). Images of the cores were exported from Zeiss Zen2011 software to enable quantification of ERK8 and VVL staining in tumour cores. To quantify the levels of ERK8 or Tn (VVL) expression in a tissue core, the images were first converted to 8-bit images using ImageJ. The area above the threshold was set for background staining (Threshold for ERK8 and VVL was 30 and DAPI was 40) and then quantified. The area of ERK8 and VVL was normalised to the total area of the core represented by nuclei (DAPI) staining. The values of each core were then normalised to the average area of the normal tissue cores.

## Additional information

### Funding

| Funder | Author |
| --- | --- |
| A*STAR | Joanne Chia, Keit Min Tham, David James Gill, Frederic A Bard |

The funders had no role in study design, data collection and interpretation, or the decision to submit the work for publication.

### Author contributions

JC, Conception and design, Acquisition of data, Analysis and interpretation of data, Drafting or revising the article, Contributed unpublished essential data or reagents; KMT, Generated cell lines for experiments; DJG, Contributed reagents used in the study; EAB-C, Analysis and interpretation of data, Drafting or revising the article; FAB, Conception and design, Analysis and interpretation of data, Drafting or revising the article

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
