## [Decision Letter]

Thank you for sending your work entitled “ERK8 is a negative regulator of O-GalNAc glycosylation and cell migration” for consideration at *eLife*. Your article has been favorably evaluated by a Senior editor and 3 reviewers, one of whom is Suzanne Pfeffer, a member of our Board of Reviewing Editors.

The Reviewing editor and the reviewers discussed their comments before we reached this decision, and the Reviewing editor has assembled the following comments to help you prepare a revised submission.

This lab has previously reported the interesting finding that several growth factors and some cancers induce the relocalization of GalNAc-transferases from the Golgi to the ER where they function to enhance GalNAc 0-glycosylation. This process leads to stimulation of cell adhesion and migration. The current study extends this work by using a RNAi screen of 948 genes involved in signal transduction to identify 12 genes that serve as negative regulators of this process, with ERK8 being the most potent and the focus of the rest of the paper. Data are presented that ERK8 is partially localized to the Golgi and must be catalytically active to retain the GalNAcTs in the Golgi. Some data are presented that ERK8 expression is downregulated in breast and lung cancers, which are associated with increased 0-GalNAc formation in the ER. Additionally it is shown that ERK8 regulated retrograde traffic of GalNAcTs is dependent on COPI.

These findings significantly extend the previous reports by identifying ERK8 as a major regulator of this process. The work has been well executed and the data are convincing that ERK8 is a regulator of GalNAcT localization.

An unexpected finding is that Erk8 depletion drives GalNAc T1 to the ER but apparently not KDELR, which is also is transported in a COP-I dependent pathway, and activated for recycling via Src family kinase activity and secretory traffic. The work would be of much greater impact if the authors could use any other molecular distinction to show that GalNAc T1 and KDEL-R really do use different activation pathways for activated retrograde transport. For example, do methods reported by Luini to enhance KDEL-R retrograde transport also activate GalNAcT1 trafficking? Are the two routes similar in sensitivity to specific tyrosine kinase inhibitors? Can they monitor the actual rate of KDEL receptor recycling rather than steady state localization or overexpress a KDEL substrate to alter the KDELR trafficking and verify that it is insensitive or less sensitive to Erk8 depletion. If they can provide any additional detail to clarify this difference, the work would represent a major finding and be precisely the kind of story we seek for *eLife*.

Regardless of the weaknesses, the identification of ERK8 as a major regulator of GalNAcT localization is a significant advance that will open up new lines of investigation and merits publication.

---

## [Author Response]

*An unexpected finding is that Erk8 depletion drives GalNAc T1 to the ER but apparently not KDELR, which is also is transported in a COP-I dependent pathway and activated for recycling via Src family kinase activity and secretory traffic. The work would be of much greater impact if the authors could use any other molecular distinction to show that GalNAc T1 and KDEL-R really do use different activation pathways for activated retrograde transport. For example, do methods reported by Luini to enhance KDEL-R retrograde transport also activate GalNAcT1 trafficking? Are the two routes similar in sensitivity to specific tyrosine kinase inhibitors? Can they monitor the actual rate of KDEL receptor recycling rather than steady state localization or overexpress a KDEL substrate to alter the KDELR trafficking and verify that it is insensitive or less sensitive to Erk8 depletion. If they can provide any additional detail to clarify this difference, the work would represent a major finding and be precisely the kind of story we seek for* eLife.

We thank the reviewers for these interesting suggestions. In this new version, we report experiments where either an active form of Src is expressed or ERK8 is depleted and both GalNac-Ts and KDEL-R are monitored simultaneously (new panels; Figure 5). Although the extent of GalNac-Ts dislocation from the Golgi is comparable in both conditions, we find that active Src expression induces a marked dispersion of KDEL-R whereas ERK8 depletion does not.

In a separate experiment, we induced a wave of cargo at the Golgi using the temperature-sensitive form of VSVG. As previously reported by Alberto Luini’s group, we observed a significant relocation of the KDEL-R, whereas, in contrast, we found that GalNac-Ts were not detectably affected (new panel; Figure 5). Together with our previous quantification, these data indicate that whereas Src controls trafficking of both KDEL-R and GalNac-Ts, ERK8 only controls GalNac-Ts traffic. Both the receptor for KDEL and the enzymes use the COPI coat machinery but their traffic is activated through different pathways: rate of secretion for KDEL-R; growth factors and events involved in cancerous transformation for GalNac-Ts.